# IFN-γ and TNF Induce Senescence and a Distinct Senescence-Associated Secretory Phenotype in Melanoma

**DOI:** 10.3390/cells11091514

**Published:** 2022-04-30

**Authors:** Lorenzo Homann, Maximilian Rentschler, Ellen Brenner, Katharina Böhm, Martin Röcken, Thomas Wieder

**Affiliations:** 1Department of Dermatology, University of Tuebingen, 72076 Tuebingen, Germany; maximilian.rentschler@med.uni-tuebingen.de (M.R.); ellen.brenner@med.uni-tuebingen.de (E.B.); katharina.boehm@med.uni-tuebingen.de (K.B.); martin.roecken@med.uni-tuebingen.de (M.R.); 2Institute of Physiology I, Department of Vegetative and Clinical Physiology, University of Tuebingen, 72074 Tuebingen, Germany

**Keywords:** senescence, melanoma, SASP, cell cycle inhibition, immunotherapy, interferon, tumor necrosis factor, tumor dormancy, doxorubicin, palbociclib

## Abstract

Immune checkpoint blockade (ICB) therapy is a central pillar of melanoma treatment leading to durable response rates. Important mechanisms of action of ICB therapy include disinhibition of CD4^+^ and CD8^+^ T cells. Stimulated CD4^+^ T helper 1 cells secrete the effector cytokines interferon-gamma (IFN-γ) and tumor necrosis factor alpha (TNF), which induce senescence in tumor cells. Besides being growth-arrested, senescent cells are metabolically active and secrete a large spectrum of factors, which are summarized as senescence-associated secretory phenotype (SASP). This secretome affects the tumor growth. Here, we compared the SASP of cytokine-induced senescent (CIS) cells with the SASP of therapy-induced senescent (TIS) cells. Therefore, we established in vitro models for CIS and TIS in melanoma. The human melanoma cell lines SK-MEL-28 and WM115 were treated with the cytokines IFN-γ and TNF as CIS, the chemotherapeutic agent doxorubicin, and the cell cycle inhibitor palbociclib as TIS. Then, we determined several senescence markers, i.e., growth arrest, p21 expression, and senescence-associated β-galactosidase (SA-β-gal) activity. For SASP analyses, we measured the regulation and secretion of several common SASP factors using qPCR arrays, protein arrays, and ELISA. Each treatment initiated a stable growth arrest, enhanced SA-β-gal activity, and—except palbociclib—increased the expression of p21. mRNA and protein analyses revealed that gene expression and secretion of SASP factors were severalfold stronger in CIS than in TIS. Finally, we showed that treatment with the conditioned media (CM) derived from cytokine- and palbociclib-treated cells induced senescence characteristics in melanoma cells. Thus, we conclude that senescence induction via cytokines may lead to self-sustaining senescence surveillance of melanoma.

## 1. Introduction

Malignant melanoma is an increasing health care issue as its incidence is rising continuously [1]. Before the clinical application of modern immunotherapy, treatment of stage IV melanoma mainly relied on the use of traditional chemotherapeutics such as dacarbazine and was only rarely successful [2]. Fortunately, immune checkpoint blockade (ICB) therapy has revolutionized the treatment of cutaneous melanoma [3]. ICB tremendously improved the treatment efficacy so that approximately one-third of patients with metastatic melanoma survive for more than three years [4]. Melanoma can be described as an immunologic malignancy; thus, immunotherapy—and in particular the T cell response—is of central importance in the treatment of malignant melanoma [5,6,7]. Targeting the immune checkpoints cytotoxic T-lymphocyte-associated protein 4 (CTLA-4) and programmed cell death protein 1 (PD-1) disinhibits both CD8^+^ and CD4^+^ T cells [8]. Activated CD4^+^ T cells release the effector cytokines interferon-gamma (IFN-γ) and tumor necrosis factor alpha (TNF), regardless of whether they were naturally stimulated by a physiologic immune reaction or reactivated by disinhibition during ICB [9,10]. In combination, both cytokines can induce senescence in cancer cells [10,11]. Cytokine-induced senescence (CIS) is an important mediator of cancer immune control underlying various immunotherapeutic approaches [3,10,11,12,13,14]. Still, more than 50% of patients with advanced melanoma do have tumor progression after twelve months, even with combination therapies of ICB [15]. A hallmark of melanoma is the loss of p16 function [16], which inhibits several cyclin-dependent kinases (CDKs) in the G1 phase of the cell cycle [17,18]. In this regard, CDK4/6 inhibitors such as palbociclib are used off-label in metastatic melanoma [3,19]. An important mechanism for tumor control by treatment with the therapeutic drug palbociclib is senescence induction [20], i.e., therapy-induced senescence (TIS) [21].

Cellular senescence is often described as a response program to different stressors, e.g., activation of oncogenes or mitochondrial dysfunction [22]. Senescence can also be triggered by cytokines, chemotherapeutic drugs, and cell cycle inhibitors [10,23,24,25,26]. Defined characteristics of senescent cells include a stable cell cycle arrest, deregulated metabolism, secretory activity, enhanced activity of the senescence-associated β-galactosidase (SA-β-gal), and macromolecular damage [22,27]. The growth arrest is manifested by upregulation of cell cycle inhibitors, such as p16, p21, or p27, and by reduction of cells in the S phase of the cell cycle [28]. Typically, p16 mediates a G1 [17,29], p21 a G1 or G2 [29,30,31], and p27 a G1 arrest [32]. As, in melanoma, *CDKN2A* (which encodes p16) has often sequence variants that result in loss of p16 function [12,16,33], the cell cycle arrest in senescent melanoma cells may be primarily mediated by p21, which is a downstream target of the tumor suppressor p53 [34].

Another feature of senescent cells is the release of numerous molecules into their surrounding environment, which are collectively known as the senescence-associated secretory phenotype (SASP) [35,36]. This SASP is extremely diverse and depends on the mode of senescence induction, the cell type, and the environment of the cells [35,37,38]. The SASP is mainly composed of proinflammatory cytokines and chemokines (e.g., IL-6, IL-7, and IL-8) and different growth factors (e.g., GRO, HGF, or IGFBPs) [35]. Moreover, some studies have shown that the SASP could reinforce or even induce senescence via cytokines such as IL-6 or IL-8 in an autocrine or paracrine manner [13,39,40]. As there is some overlap of typical SASP genes and IFN-stimulated genes [41], we thus hypothesized that the SASP during CIS contains higher concentrations of cytokines and chemokines than during TIS.

In the present study, we induced senescence in two human melanoma cell lines, SK-MEL-28 and WM115, with an established cytokine cocktail of IFN-γ and TNF, the chemotherapeutic agent doxorubicin, and the CDK4/6 inhibitor palbociclib. Senescence markers were measured at the time of senescence induction and 48 h later to analyze the individual dynamics of the different cellular phenotypes and also to validate that the vast majority of cells were senescent at the time of SASP analyses. Then, the gene expression of 84 cytokines and chemokines was determined 48 h after the removal of the respective senescence inducer to ensure that no acute response to the treatment was measured. At the same time, the supernatants were analyzed, and 105 different factors commonly associated with the SASP were measured on the protein level. Finally, we treated SK-MEL-28 and WM115 melanoma cells with the different conditioned media (CM) to determine if the respective SASP could induce senescence in melanoma cells.

## 2. Materials and Methods

### 2.1. Cell Culture

Human melanoma cell lines SK-MEL-28 and WM115 were cultured in RPMI 1640 medium supplemented with 10% fetal calf serum (FCS), L-glutamine, MEM amino acids, penicillin/streptomycin, HEPES buffer, and sodium pyruvate (all obtained from Biochrom AG, Berlin, Germany). The SK-MEL-28 cell line was kindly provided by B. Schittek (University of Tuebingen, Tuebingen, Germany) and the WM115 cell line by T. Feuchtinger (University of Tuebingen, Tuebingen, Germany). Recombinant human (rh) IFN-γ and rh TNF [10] (R&D Systems, Minneapolis, MN, USA) were used at concentrations of 125 ng/mL for IFN-γ and 12.5 ng/mL for TNF. Doxorubicin hydrochloride was dissolved in dimethyl sulfoxide (DMSO; both from Sigma Aldrich, St. Louis, MO, USA) [42] and used at concentrations of 50 nM for the SK-MEL-28 cell line and 25 nM for the WM115 cell line. Palbociclib (Selleckchem, Houston, TX, USA) was dissolved in DMSO [43] and used at a concentration of 8 μM for both cell lines. In general, medium was used as a control for the cytokine-treated cells, while medium supplemented with 0.12% DMSO was used for the doxorubicin- and palbociclib-treated cells. 24 h before treatment, cells from both cell lines were seeded at a density of 10^4^ cells/cm^2^. Then, they were treated with IFN-γ and TNF for 96 h, palbociclib for 96 h, or doxorubicin for 24 h and subsequently 72 h with medium in the absence of the drug (total induction time of CIS is 96 h [10]; to allow comparability, the induction time for TIS was also defined as 96 h).

### 2.2. Generation of Conditioned Media (CM)

The supernatants of senescent cells collected 48 h after the end of the treatment and removal of the senescence inducers are defined as CM. The different CM were used for the analysis of SASP factors and as a 96 h treatment for naïve, non-senescent melanoma cells. To obtain the CM, all culture dishes were washed thoroughly with PBS after the removal of the senescence triggers. Subsequently, the cells were replenished with fresh medium and cultured for another 48 h. Then, cell culture supernatants from each condition were collected.

### 2.3. Senescence-Associated β-Galactosidase (SA-β-Gal) Staining

The activity of the SA-β-gal was determined using the Senescence Detection Kit (Assay Genie, Dublin, Ireland) as described [10]. After treatment, cells were washed with PBS, then fixed for 10 min at room temperature using the kit’s formaldehyde- and glutaraldehyde-containing fixative solution. The X-gal substrate was dissolved in N,N-dimethylformamide, and finally, cells were stained with X-gal and the provided staining solution at pH 6.0 for 16 h at 37 °C in an incubator (Heraeus, Hanau, Germany). Subsequently, cells were washed with PBS and stained with 4′,6-diamidino-2-phenylindole (DAPI; Invitrogen, Carlsbad, CA, USA) to allow the determination of the total cell number. SA-β-gal-positive cells (blue cellular staining) and DAPI-positive nuclei were counted using an Axiovert 200 microscope (Zeiss, Oberkochen, Germany) and the ImageJ software V. 1.53a (Wayne Rasband, Bethesda, MD, USA). For each condition, three replicates were evaluated, and only fields showing at least 100 cells were analyzed. Finally, the number of SA-β-gal-positive cells was calculated in % of the total cell population.

### 2.4. Lactate Dehydrogenase (LDH) Release Assay

To measure the treatment-induced LDH release, the CyQUANT LDH Cytotoxicity Assay (Thermo Fisher Scientific, Waltham, MA, USA) was performed according to the manufacturer’s protocol. Cells were seeded onto 96-well plates with 1500 cells/well. For each condition, six replicates were treated as described in Section 2.1 and additionally with 0.1% Triton X-100 (Roth, Karlsruhe, Germany) that served as a positive control. Lysis buffer was added to three of the six wells for 45 min to reach maximum LDH release, and the same amount of pyrogen-free water (Ampuwa; Fresenius, Bad Homburg, Germany) was added to the three remaining wells, respectively. Then, the supernatants were transferred to new 96-well plates before adding the reaction mixture for 30 min. Subsequently, the stop solution was added, and the change in color of each well was analyzed using a Multiskan Ex microplate reader (Thermo Fisher Scientific). After subtracting the blank value from each reading, the absorbance of the sham-treated supernatants was divided by the absorbance of the lysis buffer-treated supernatants to calculate the LDH release in %. Values over 100% are indicated as 100% and values below 0% as 0%.

### 2.5. In Vitro Growth Arrest Assay

The growth arrest assay was performed as described previously [10]. Briefly, cells were seeded at a density of 10^4^ cells/cm^2^ and treated with medium, cytokines, DMSO, doxorubicin, and palbociclib (for details, see Section 2.1). After 96 h, cells were washed, trypsinized, and counted. The number of viable cells was determined under a Zeiss Primovert microscope (Zeiss) using trypan blue solution (Gibco, Waltham, MA, USA) and a Neubauer chamber (Karl Hecht, Sondheim von der Rhön, Germany). Then, cells were seeded again at a density of 10^4^ cells/cm^2^ and cultured in fresh medium for 3 days. Afterward, the number of cells was determined as described above. Subsequently, cells were again seeded and cultured for 3 additional days, before the final cell number was assessed.

### 2.6. Western Blot

Western blot analysis was performed as described [44]. Cell lysates were prepared after the respective treatment using a cell scraper and RIPA lysis buffer containing 50 mM Tris-HCl (pH 7.5), 150 mM NaCl, 1% Triton X-100, 0.5% sodium dodecyl sulfate (SDS), 1 mM Na_3_VO_4_, 0.4% β-mercaptoethanol, and a protease inhibitor cocktail (cOmplete; Roche, Basel, Switzerland). The protein concentration was determined using the Pierce bicinchoninic acid (BCA) Protein Assay Kit (Thermo Fisher Scientific). Proteins were resolved using sodium dodecyl sulfate-polyacrylamide gel electrophoresis (SDS-PAGE) and then transferred onto an Immobilon FL polyvinylidene difluoride (PVDF) membrane (Merck, Darmstadt, Germany). Subsequently, the membranes were blocked using Intercept Blocking Buffer (Li-Cor, Lincoln, NE, USA). Protein targets of interest were detected on the membranes after overnight incubation with primary antibodies directed against p16 ^Ink4a^ (1:1000; D7C1M, Cell Signaling, Danvers, MA, USA), p21 ^Waf1/Cip1^ (1:1000; 12D1, Cell Signaling), p27 ^Kip1^ (1:1000; D69C12, Cell Signaling), or p53 (1:500; DO-1, Santa Cruz, Dallas, TX, USA), each in combination with an antibody detecting β-actin (1:5000; clone C4, Millipore, Burlington, MA, USA, or, respectively, 1:1000; 13E5, Cell Signaling). Afterward, membranes were washed using Tris-buffered saline supplemented with Tween 20 (Roth). The detection was then performed with fluorescent dye-labeled secondary antibodies (goat anti-rabbit IRDye 800CW (1:15,000) and goat anti-mouse IRDye 680RD (1:15,000), both from Li-Cor) and the Li-Cor Odyssey SA Imaging System. The scanned membranes were analyzed using the Image Studio Lite software (Li-Cor), and the ratio between each target (p16, p21, p27, or p53) and the respective reference (β-actin) was calculated. Finally, the obtained values were normalized to the 0 h medium control, which was set as 1. HeLa cell lysates were used as positive controls.

### 2.7. Cell Cycle Analysis

The APC BrdU Flow Kit (Becton Dickinson, Franklin Lakes, NJ, USA) was used for flow cytometric analysis of the cell cycle according to the manufacturer’s protocol. After the respective treatment (either cytokines, drugs, or controls), BrdU solution was added for another 3 h into the medium of the cells (10 µL of 1 mM BrdU per mL culture medium). Then, both the adherent as well as the detached cells in the supernatant were collected and counted. A total of 5 × 10^5^ cells/well were placed into a 96-deep-well plate, fixed, permeablized, and stained according to the manufacturer’s protocol. Flow cytometric measurements were performed on an LSRII cytometer in combination with the FACSDiva software V. 9.0, and the derived data were analyzed using the FlowJo software V. 10.8.0 (all from Becton Dickinson).

### 2.8. RNA Isolation and cDNA Synthesis

RNA isolation and cDNA synthesis were performed using the NucleoSpin RNA Plus Kit (Macherey-Nagel, Düren, Germany) and iScript cDNA Synthesis Kit (Bio-Rad, Hercules, CA, USA) according to the protocol provided by the respective manufacturer. After treatment, cells were trypsinized and resuspended in lysis buffer, and the lysates were then purified. Finally, the RNA concentration and the quality markers OD^260/230^ and OD^260/280^ were determined using a BioPhotometer 6131 Spectrometer (Eppendorf, Hamburg, Germany).

### 2.9. qPCR Arrays

The isolated mRNA was reversely transcribed to cDNA and further analyzed using human PrimePCR Arrays (Bio-Rad) with a predesigned target panel for cytokines and chemokines (SAB target list) according to the manufacturer’s conditions with the following settings in a LightCycler 480 system (Roche): 95 °C for 2 min, 40× (95 °C for 5 s, 60 °C for 30 s), melting curve from 95 °C to 65 °C with 0.1 °C/s.

Array data were analyzed using the PrimePCR Analysis Software (Bio-Rad). The expression of target genes was normalized to the expression of the reference genes *GAPDH*, *HPRT1*, and *TBP*.

### 2.10. Proteome Profiler Arrays

The secreted factors in the supernatants of senescent cells were analyzed using antibody-based Proteome Profiler Human XL Cytokine Array Kits (R&D Systems) essentially as described in the manufacturer’s protocol. After blocking with block buffer, the membranes were incubated with the supernatants overnight. Then, the array membranes were washed and incubated with the detection antibody cocktail. IRDye 800CW Streptavidin (1:2000; Li-Cor) was used as an alternative detection method. The array membranes were then analyzed using the Odyssey SA Imaging System in combination with the Image Studio Lite software (both from Li-Cor).

### 2.11. Enzyme-Linked Immunosorbent Assay (ELISA)

Individual factors in the supernatants of senescent cells were measured by ELISA as described [45]. For this, DuoSet ELISA Kits for IL-1β, IL-6, IL-8, and IL-12 (all from R&D Systems) were used according to the manufacturer’s protocol; the ELISA kit for IL-12 is able to detect not only the whole protein IL-12 p70 but also both subunits p35 and p40. In brief, 96-well plates were coated with the respective capture antibody overnight. The next day, the plates were washed, blocked, and washed again, before the samples and protein standards were added for analysis. Afterward, the detection antibody in combination with Streptavidin-HRP and the substrate solution was used to measure the change in color of the individual wells with a Multiskan Ex microplate reader (Thermo Fisher Scientific). The concentration of the target protein was then calculated by subtracting the background measurement at a reference wavelength (540 nm) from the actual values at the test wavelength (450 nm) and the use of a four-parameter logistic regression.

### 2.12. Statistical Analysis

GraphPad Prism 8 (GraphPad Software Inc., San Diego, CA, USA) was used for statistical analysis. Data are expressed as arithmetic means ± standard deviation (s.d.), if not stated otherwise. Unpaired Student’s *t*-test and one- or two-way analysis of variance (ANOVA) with Tukey’s multiple comparisons test were performed when appropriate. An asterisk (*) indicates a *p*-value < 0.05, which was considered statistically significant.

## 3. Results

### 3.1. Senescence Induction in Melanoma Cells with Cytokines, Doxorubicin, and Palbociclib

First of all, we established senescence induction in the human melanoma cell lines SK-MEL-28 and WM115 using the cytokine cocktail IFN-γ and TNF and the drugs doxorubicin and palbociclib. The treatment with IFN-γ and TNF, doxorubicin, and palbociclib strongly enhanced the activity of SA-β-gal after 96 h in both the SK-MEL-28 (Figure 1a) and the WM115 (Figure 1b) cell line. The percentage of SA-β-gal-positive cells reached 30–60% of the total population for both cell lines and all three inducers, whereas the SA-β-gal-positive cells in the controls did not exceed 10%. Importantly, the SA-β-gal activity remained increased for another 48 h after the end of treatment (Figure 1c). The treated cells also showed decreased cell density and the typical senescence-associated cellular phenotype, as they adopted a flattened and enlarged morphology (Figure 1a,b). Furthermore, some senescent cells also displayed polyploidy or multinucleation, which was particularly visible after treatment with doxorubicin (Figure 1a,b, black and white arrows).

A key feature of senescent cells is the formation of a stable growth arrest. We thus determined the cell numbers of SK-MEL-28 (Figure 2a, Table 1) and WM115 (Figure 2b, Table 1) melanoma cells at the end of each treatment and for two passages after withdrawal of the senescence inducers. IFN-γ and TNF, doxorubicin, and palbociclib treatment inhibited the growth of melanoma cells during the treatment period. More importantly, all three treatment regimens induced a stable growth arrest as the cells did not restart exponential growth during the two passages after the removal of the respective senescence inducers (Figure 2a,b, Table 1).

Moreover, we determined the lactate dehydrogenase (LDH) release into the supernatants induced by each treatment after the 96 h senescence induction. Damage to the cell membrane leads to the release of the cytosolic enzyme LDH, which is associated with cytotoxicity but not necessarily with an inevitable cell death [46]. Regarding the WM115 cells, none of the treatments resulted in increased levels of LDH release (Figure 2c). Elevated LDH levels were only observed for the cytokine- and doxorubicin-treated SK-MEL-28 cells, but the increase did not exceed 15% (Figure 2c). In contrast, treatment with palbociclib in SK-MEL-28 cells did not lead to an increased LDH release. Altogether, only little to no LDH release was induced by the different treatment regimens.

### 3.2. Stable Induction of p21 in Melanoma Cells after Treatment with Cytokines and Doxorubicin, but Not with Palbociclib

After establishing senescence with cytokines (CIS) and chemical compounds (TIS) in SK-MEL-28 and WM115 cells, we determined which cell cycle regulatory proteins mediated the stable growth arrest. The role of p16 and p21 in senescence has been described extensively, but p27 is also capable of inhibiting the proliferation of cells, even though it is rather associated with dormancy than senescence [22,32]. First, we investigated the expression of p16. None of the senescence inducers altered the expression of p16 in the SK-MEL-28 cell line, while this protein was not detectable in the p16-deficient cell line WM115 (Figure 3a). In contrast, the cytokine- and doxorubicin-treated cells showed a stable induction of p21 while palbociclib did not increase p21 expression (Figure 3b). None of the senescence inducers increased the expression of p27 as compared to the respective controls (Figure 3c). The expression of p53, which acts upstream of p21, was reduced in the SK-MEL-28 cells at 144 h following treatment with cytokines and palbociclib as compared to the respective controls (Figure 3d). However, doxorubicin-treated WM115 cells showed a stable induction of p53 at both time points. Moreover, the cytokine treatment led to an enhanced expression of p53 in the WM115 cells after 144 h as compared to the medium control. Therefore, the induction of p21 is not necessarily accompanied by an increase in the protein level of total p53 (Figure 3d).

Taken together, the cell cycle arrest in the two human melanoma cell lines SK-MEL-28 and WM115 is either mediated indirectly by the induction of p21, which inhibits various CDKs, as in the case of cytokines and doxorubicin, or directly by the synthetic CDK4/6 inhibitor palbociclib.

### 3.3. Stable Cell Cycle Arrest Following Treatment with Cytokines, Doxorubicin, and Palbociclib

Next, we further characterized the type of cell cycle arrest in CIS and TIS. Since p21 acted as the key cell cycle regulator, the melanoma cells may arrest in the G1 or the G2 phase [29,30,31]. Treatment with cytokines, doxorubicin, and palbociclib diminished the proportion of SK-MEL-28 (Figure 4a–c, Table A1) and WM115 (Figure A1a–c, Table A1) melanoma cells in the S phase, thereby corroborating the successful induction of a growth arrest induced by all three senescence triggers. Moreover, the subG1 phase increased in response to cytokine and doxorubicin treatment. After 96 h and 144 h, the majority of cytokine-treated melanoma cells remained in the G1 phase, while the percentage of cells in both the subG1 and G2 phase increased (Figure 4b,c, Figure A1b,c). Senescence induction with the genotoxic drug doxorubicin was associated with a notable fraction of cells in the subG1 phase, which represents apoptosis (Figure 4b,c, Figure A1b,c). Palbociclib, on the other hand, had no such effect as compared to the DMSO control (Figure 4d (SK-MEL-28), Figure 4e (WM115)). Consistent with the findings shown in Figure 1a,b, doxorubicin treatment resulted in considerably more polyploid cells (>4n) as compared to the treatment with cytokines and palbociclib (Figure 4d (SK-MEL-28), Figure 4e (WM115)). Taken together, senescent melanoma cells arrested in different phases of the cell cycle, depending on the senescence inducer.

To recapitulate, all three senescence inducers enhanced SA-β-gal activity (Figure 1) and induced a stable growth arrest not only at the end of the 96 h treatment, but also 48 h later (Figure 2 and Figure 4). Although only approx. 35% of the cytokine-treated cells were SA-β-gal-positive (Figure 1), the vast majority of cells (>95%) stopped proliferating (Figure 4), which indicates a stable senescence induction. Thus, the secretome of senescent cells and its factors can be measured by RNA and protein analyses 48 h after removal of the senescence inducers (time point 144 h for all three conditions).

### 3.4. Cytokine-Induced Senescence in Melanoma Cells Leads to a Pronounced Secretion of Cytokines and Chemokines

After establishing and characterizing the inducer-dependent phenotypes of senescent melanoma cells, we compared the SASP after treatment with cytokines (CIS) and therapeutic drugs (TIS). To decipher the secretome of senescent cells rather than the direct effects of cytokine or drug treatment, the following analyses were performed 48 h after the end of each treatment and, thus, after the removal of the senescence inducers. qPCR array analysis revealed that all treatment regimens induced the gene expression of cytokines and chemokines in both melanoma cell lines (Figure A2a,b). However, the expression of SASP-related genes, mainly of proinflammatory factors, was increased to a much higher extent in CIS than in TIS. Especially the expression of interleukins, e.g., *IL1B* and *IL8*, and chemokines, e.g., *CXCL10* and *CXCL11*, was maintained at high levels in CIS.

Next, we performed an array-based screening for cytokines and chemokines in the supernatants of senescent cells, again 48 h after removal of the inducers. Similar to the RNA-based data, the levels of secreted cytokines and chemokines in the supernatants were severalfold higher in the SASP of cytokine-treated SK-MEL-28 and WM115 cells (CIS) as compared to the SASP of the respective therapeutic drug-treated melanoma cells (TIS; Figure 5a,b). The protein levels were normalized to the respective protein content of the supernatant produced by the medium control.

To validate the array-based data, we performed single ELISA analyses of the prominent SASP factors IL-1β, IL-6, IL-8, and IL-12 [13,35,40,47,48,49]. As the supernatants which were analyzed in Figure 5a–d were obtained from cell populations of extremely different cell numbers (see also Figure 2a,b and Table 1), we also determined the cell number of each population after obtaining the supernatants and normalized the protein levels measured by ELISA to 500,000 cells. In these experiments, we observed a significantly increased secretion of IL-1β, IL-6, and IL-8 in cytokine-treated SK-MEL-28 and WM115 cells as compared to the appropriate controls (Figure 5c,d). Therapeutic drug-treated SK-MEL-28 and WM115 cells also showed IL-1β and IL-6 secretion, but it was less pronounced than in CIS cells. Especially the secretion of IL-1β from CIS cells was 5- to 10-fold higher than the IL-1β secretion from doxorubicin-treated cells, and even 30- to 80-fold higher than the secretion from palbociclib-treated cells (Figure 5c,d). The data thus confirmed the prominent appearance of IL-1β in the SASP of cytokine-induced senescent melanoma cells, which was already indicated in the qPCR and Proteome Profiler arrays (Figure 5a,b and Figure A2a,b).

Furthermore, no secretion of IL-12 (p70) was observed, although the ELISA is able to specifically detect the subunits and the whole protein. However, this was expected since only the expression of the *IL12A* gene encoding the IL-12 p35 subunit was upregulated (Figure A2a,b), and the complete IL-12 p70 protein was not secreted by melanoma cells.

To assess whether the different SASPs could induce senescence, SK-MEL-28 and WM115 melanoma cells were treated with the conditioned media (CM) derived from the different treatments and controls for 96 h. Then, SA-β-gal activity staining was performed (Figure 6a,b), and the percentage of SA-β-gal-positive cells was calculated (Figure 6c). Treatment with I+T-CM and Palbo-CM induced SA-β-gal activity and a senescence-associated change in the cellular morphology, as the cells appeared larger and flattened (Figure 6a,b). In contrast, we observed no effects on the SA-β-gal activity following treatment with Doxo-CM (Figure 6a,b).

Taken together, melanoma cells secreted a variety of proinflammatory factors, mainly cytokines, such as IL-1β and IL-8, and chemokines, such as CXCL10, CXCL11, and CCL20, during CIS, while during TIS, melanoma cells produced lower levels of SASP proteins.

## 4. Discussion

This comparative study demonstrated that CIS drives melanoma cells to secrete considerably more cytokines and chemokines than melanoma cells during TIS. Moreover, we showed that senescent melanoma cells arrest in different phases of the cell cycle, depending on the inducer. Additionally, we observed that p21 is the main cell cycle inhibitor most closely associated with the therapeutic regimens of the tested melanoma cell lines.

CIS can be observed independently of immunotherapy [10,50]. For example, IFN-γ and TNF are also released during a natural immune response involving T helper 1 (T_H_1) cells (for instance, against tumor cells), although in lower concentrations [10,12], or against macrophages infected with *Mycobacterium tuberculosis* without disinhibition by ICB [51]. Furthermore, NK cells secrete the senescence-inducing cytokines IFN-γ and TNF after stimulation with type I IFNs [52,53].

Regarding melanoma treatment, CIS evidently plays a crucial role in the context of immunotherapy. The relationship between immunotherapy and CIS has been explored in different experimental settings including metastatic melanoma [10,12]; in the latter study, significantly more fully inactivating mutations of senescence-inducing cell cycle genes were observed in melanoma of patients who did not respond to ICB as compared to responder patients [12]. As the regulation of the respective cell cycle pathways is interferon-dependent, successful induction of CIS is required for tumor immune control in the context of ICB [12].

In melanoma cells, the function of p16 (encoded by *CDKN2A*) or the p16 pathway is often disrupted [16]. The SK-MEL-28 cell line expresses wildtype *CDKN2A* but carries a *CDK4* point mutation [33], which encodes a downstream target of p16. The WM115 cell line, on the other hand, expresses wildtype *CDK4* but is *CDKN2A*-deficient [16]. However, wildtype *CDKN1A* and *CDKN1B*, which encode for p21 and p27, are expressed in both cell lines [54,55,56,57]. Altogether, we assumed that the cell cycle arrest would be mediated by either p21 or p27, and we expected an enhanced expression of one or both proteins acting upstream of CDK4/6 in all treatment regimens.

Upregulation of p21 was only measured in cytokine- or doxorubicin- but not palbociclib-treated senescent SK-MEL-28 and WM115 melanoma cells. Palbociclib inhibits CDK4/6, which in turn inhibits p21 [58]; thus, we assumed disinhibition and therefore an upregulation of p21 expression, which has been shown in some studies [59,60]. Otherwise, Leontieva et al. proposed that CDK4/6 inhibitors might substitute for p21 in senescence [61]. Our data argue in favor of this idea as palbociclib-treated senescent melanoma cells did not express p16, p21, or p27 in high amounts. Furthermore, lysosomal trapping of palbociclib and a subsequent paracrine release after the treatment phase could account for the lack of p21 upregulation [62]. Regarding p53, stable induction of this tumor suppressor was observed only in doxorubicin-treated WM115 cells, even though p21 was upregulated in both cell lines in response to cytokine and doxorubicin treatment. These results suggest a p53-independent regulation of p21 in SK-MEL-28 cells, which has been observed previously in other contexts [63,64].

The SASP is often described as an undesirable side effect of senescence [24]. However, its role in influencing neighboring cells in the tumor microenvironment (i.e., cancer, stromal, or immune cells) is as multifaceted as its composition [50]. The SASP can promote tumor growth, induce epithelial-mesenchymal transition, or enhance the motility of neighboring cells, but it can also reinforce the senescent state in an autocrine or paracrine manner, activate and attract immune cells that recognize and destroy senescent cells, and induce tissue repair [24,37,65]. By releasing many cytokines and chemokines, the SASP contributes to an inflammatory environment. It must also be considered that the effects of an inflammatory milieu partly depend on its duration. In this context, a therapeutic induction of short-term inflammation, e.g., by adoptive transfer of immune cells [10,12], efficiently controls cancer growth and can be considered anti-tumoral.

In the following sections of the discussion, the effects of individual SASP components are described, which have been observed in various settings. Since effects on immune cells or direct links between ICB and senescence were not analyzed in the context of the present study, the focus relies on a detailed description of the identified SASPs in melanoma that are established in response to different therapeutic regimens.

The most prominent components of the SASP in cytokine-treated melanoma cells were IL-1β, IL-6, and IL-8. On one hand, this secretome could be deleterious, as IL-1β might increase vascular permeability [66], and IL-6 could inhibit dendritic cells (DCs) [67], support tumor cell migration [68], and, just like IL-8, is able to promote tumor growth [69,70]. Moreover, melanoma patients with high serum IL-8 levels did not respond as well to ICB therapy as patients with low serum levels of IL-8 [71]. On the other hand, this secretome could be beneficial, as IL-1β was shown to inhibit melanoma growth in vivo [72,73], and IL-6, as well as IL-8, could not only induce but also reinforce senescence in the MCF-7 breast cancer cell line [40,74,75]. Furthermore, IL-1β and other factors, which were also strongly elevated in CIS (Figure 5a,b), are important factors for inflammasome activation, a process for which both tumor-promoting and -inhibiting effects have been described [70,76]. The secretion of CCL2, which was induced in SK-MEL-28 melanoma cells (Figure 5a), contributes to the clearance of senescent tumor cells in hepatocellular carcinoma through the recruitment of myeloid cells [77]. Moreover, our data revealed that the SASP of cytokine-treated melanoma cells contains high amounts of anti-angiogenic chemokines such as CXCL10 or CXCL11 (Figure 5a,b), thereby pointing to an acute, anti-tumoral effect of CIS that has already been described [10].

Regarding the effects of the different CM on tumor cells, we found that both I+T-CM and Palbo-CM induced senescence-associated characteristics, whereas Doxo-CM had no such effect. This may be due to certain factors released after treatment with cytokines and palbociclib, but not after treatment with doxorubicin. In accordance with a study that reported trapping of palbociclib during the treatment phase within the melanoma cells [62], we suggest that this mechanism explains both the lack of p21 upregulation (Figure 3b) as well as the induction of SA-β-gal activity after treatment with Palbo-CM (Figure 6). Thus, cytokine-treated and palbociclib-treated senescent melanoma cells may induce senescence characteristics in tumor cells by different mechanisms.

IFN-γ and TNF are the effector cytokines of T_H_1 cells that are disinhibited by ICB. In the context of ICB and T_H_1 cell cytokines, a recent study showed that tumor-infiltrating DCs express an increased IL-12/IL-10 ratio and more co-stimulatory molecules [78]. This promotes a T cell response as DCs can present antigens more efficiently, leading to an enhanced secretion of the senescence-inducing effector cytokines IFN-γ and TNF by T_H_1 cells.

In summary, the SASP of cytokine-treated melanoma cells could lead to self-sustaining senescence surveillance by the immune system. After targeting senescent cells, NK cells produce IFN-γ and TNF and thereby attract and activate macrophages [52,79,80]; this cytokine secretion might also lead to paracrine senescence induction. However, the SASP could also aggravate the disease if the proinflammatory factors are released continuously. The timely immunoclearence of SASP-producing senescent tumor cells, however, stops the release of proinflammatory factors [79]. By this mechanism, the immune system tilts the balance toward the beneficial effects of the inflammatory milieu [24]. Not only immune cells but also senolytic drugs such as BCL-2 inhibitors can selectively eliminate senescent cells and, thus, blunt the potentially tumor growth-promoting SASP [81,82]. A “one-two punch” therapeutic approach, where tumor patients are first treated with senescence-inducing agents and subsequently with senolytic drugs, may improve the treatment of cancer while simultaneously mitigating side effects caused by the chronic long-term secretion of SASP factors [83,84,85]. Considering the different and, to some extent, contradictory effects of the discussed cytokines and chemokines on the tumor microenvironment, further in vivo studies are needed to ascertain that the cancer immune control is mediated by the robust secretion of the proinflammatory SASP factors which is triggered by CIS.

## 5. Conclusions

The upregulation of p21 is induced by treatment with cytokines and doxorubicin, but not with palbociclib. Palbociclib may be trapped in vesicles and released later by melanoma cells into the cell culture medium [62], substituting for the function of p21.

The SASP generated by CIS melanoma cells shows an enhanced proinflammatory profile as compared to the SASP of TIS melanoma cells.

The CM produced by senescent melanoma cells following treatment with cytokines and palbociclib, but not doxorubicin, may induce senescence in melanoma cells by two different mechanisms: the CM generated by cytokine-treated melanoma cells contains senescence-inducing SASP factors, whereas the CM derived from palbociclib-treated melanoma cells may still contain an amount of this cell cycle inhibitor stored in vesicles.

## Figures and Tables

**Figure 1 cells-11-01514-f001:**
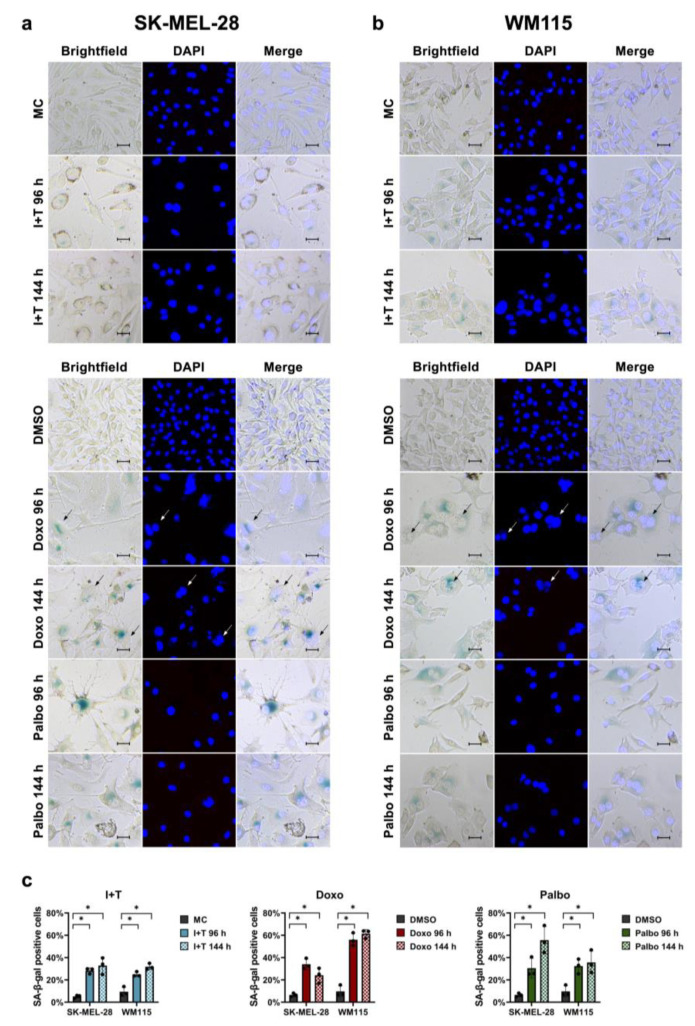
Induction of SA-β-gal activity after treatment with IFN-γ and TNF (I+T), doxorubicin (Doxo), and palbociclib (Palbo). (**a**,**b**) Representative images of SA-β-gal and DAPI staining in SK-MEL-28 (**a**) and WM115 (**b**) cells are shown after the 96 h treatment and 48 h after the end of the treatment (144 h). (**c**) Quantification of SA-β-gal-positive cells. MC, medium control after 96 h. DMSO, vehicle control after 96 h; same values for DMSO are shown in (**c**), as this control was used for both treatments with Doxo and Palbo. Bars = 50 μm. Black and white arrows highlight polyploid or multinucleated cells. The percentages of SA-β-gal-positive cells are from three independent experiments (*n* = 3); unpaired Student’s *t*-test; ns, not significant; mean ± s.d.; an asterisk (*) indicates a significant difference from the respective controls with *p* < 0.05.

**Figure 2 cells-11-01514-f002:**
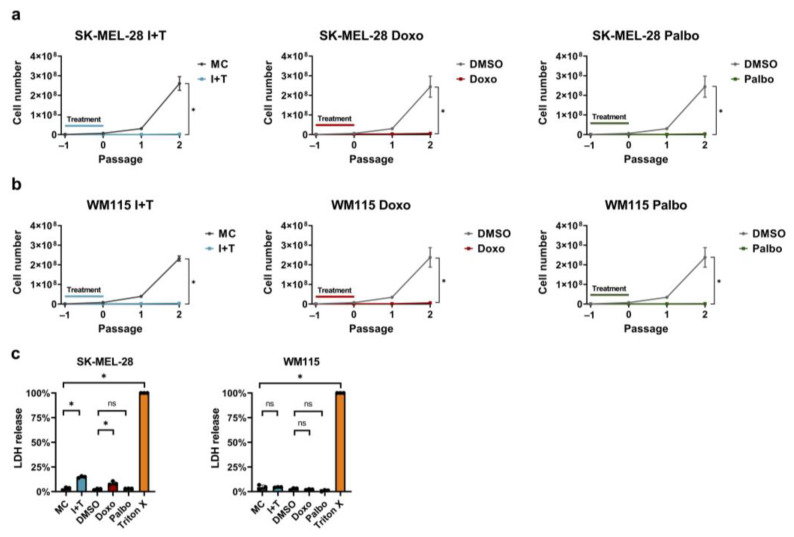
Treatment with IFN-γ and TNF, doxorubicin, and palbociclib induces a stable growth arrest and little to no LDH release. (**a**,**b**) Growth curves of SK-MEL-28 (**a**) and WM115 (**b**) melanoma cells treated with IFN-γ and TNF (I+T), doxorubicin (Doxo), and palbociclib (Palbo). MC, medium control. DMSO, vehicle control; same values for the DMSO control are shown in (**a**,**b**), as the same DMSO control was used for both treatments with Doxo and Palbo. The exact cell numbers for each condition are shown in Table 1. (**c**) Relative LDH release of each treatment was assessed for SK-MEL-28 (left panel) and WM115 (right panel) melanoma cells by the induced release of LDH after 96 h in relation to the maximum release of LDH for the respective condition. (**a**–**c**) Cell numbers and the results from the LDH release assay are from three independent experiments (*n* = 3); unpaired Student’s *t*-test; mean ± s.d.; an asterisk (*) indicates a significant difference from the respective controls with *p* < 0.05.

**Figure 3 cells-11-01514-f003:**
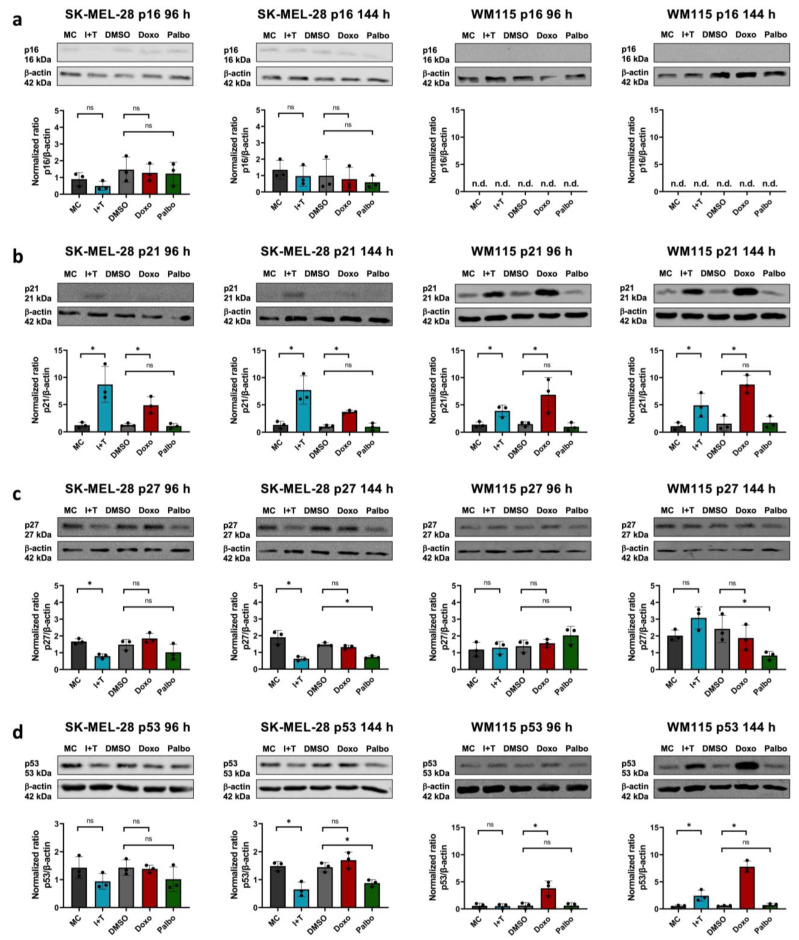
Induction of the cell cycle inhibitor p21 in melanoma cells after cytokine and doxorubicin treatment. Original Western blots (upper row) of p16 (**a**), p21 (**b**), p27 (**c**), and p53 (**d**) in SK-MEL-28 (first and second column) or WM115 (third and fourth column) melanoma cells treated with IFN-γ and TNF (I+T), doxorubicin (Doxo), and palbociclib (Palbo) for 96 h or 48 h after the end of treatment (144 h) as indicated. MC, medium control. DMSO, vehicle control. Lower panels show the semiquantitative densitometric analysis of p16 (**a**), p21 (**b**), p27 (**c**), and p53 (**d**) expression (upper row) and normalization to β-actin (lower row). The respective ratio of each control and treatment was normalized to the medium control at 0 h, which was set as 1 (data not shown). Results are from three independent experiments (*n* = 3); n.d., not detectable; ns, not significant; unpaired Student’s *t*-test; mean ± s.d.; an asterisk (*) indicates a significant difference from the respective controls with *p* < 0.05.

**Figure 4 cells-11-01514-f004:**
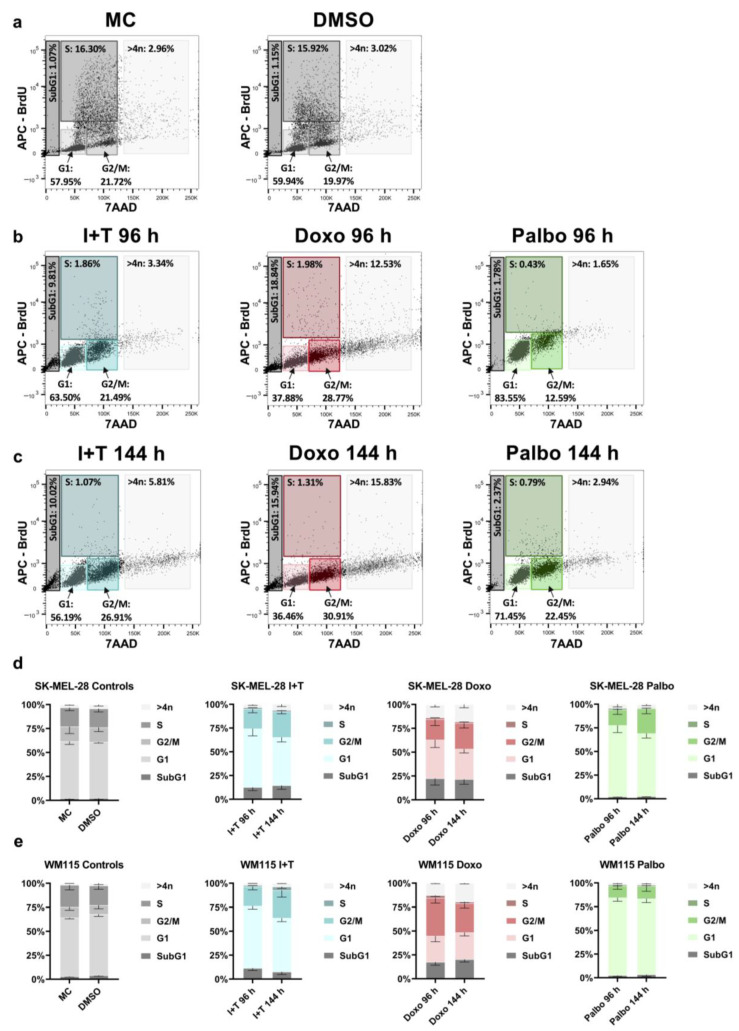
Differential cell cycle arrest induced by cytokines, doxorubicin, and palbociclib. (**a–c**) Representative plots depicting the distribution of SK-MEL-28 cells in different phases of the cell cycle (subG1, G1, S, G2/M, >4n) upon treatment with IFN-γ and TNF (I+T), doxorubicin (Doxo), and palbociclib (Palbo), and respective controls as indicated. MC, medium control. DMSO, vehicle control. Flow cytometric analysis was performed after the 96 h treatment (**a**,**b**) and 48 h after the end of treatment (144 h; (**c**)). (**d**,**e**) Quantification of cell cycle phases (subG1, G1, S, G2/M, >4n) in SK-MEL-28 (**d**) and WM115 (**e**) melanoma cells after CIS or TIS. (**a–c**) Representative images from one of three experiments are shown. (**d**,**e**) Results are from three independent experiments (*n* = 3) and show the mean ± s.d.; statistical analysis by two-way ANOVA shows a significant difference (*p* < 0.05) between the S phase of each treatment at both time points and the respective controls (*p*-values are indicated in Table A1).

**Figure 5 cells-11-01514-f005:**
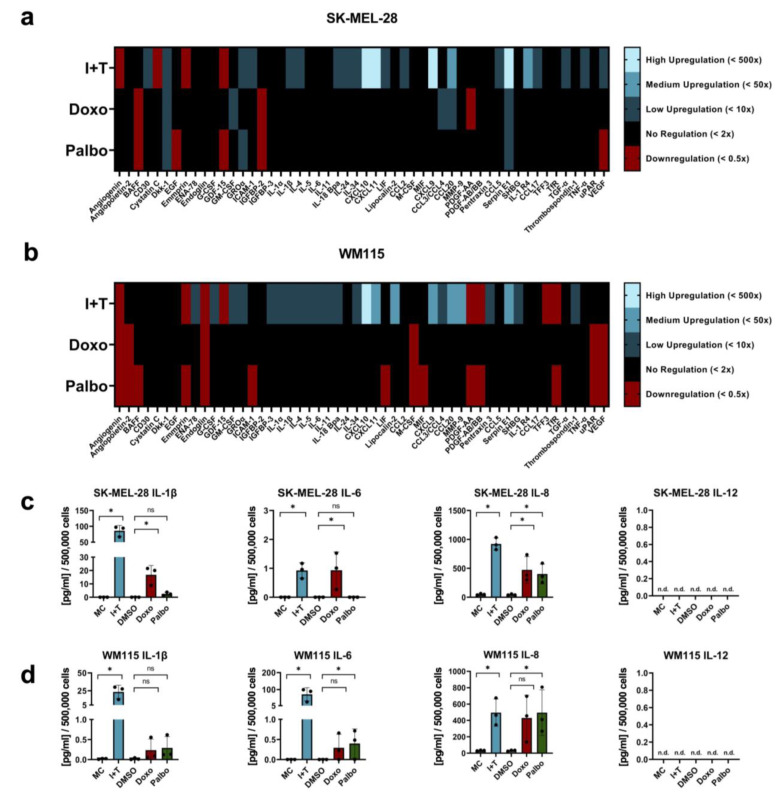
SASP analysis of cytokine- and therapy-induced senescent melanoma cells. (**a**,**b**) Secretion of cytokines and chemokines by senescent SK-MEL-28 (**a**) and WM115 (**b**) melanoma cells as determined by Proteome Profiler arrays. Supernatants were analyzed 48 h after the end of treatment (i.e., 144 h); the number of cells in each control and treatment whose supernatants were analyzed is indicated in Table A2. Densitometric measurements of proteins in the supernatants derived from cytokine-, doxorubicin-, and palbociclib-treated cells were normalized to the medium control. (**c,d**) Quantification of IL-1β, IL-6, IL-8, and IL-12 levels in the supernatants of senescent SK-MEL-28 (**c**) and WM115 (**d**) melanoma cells as measured by ELISA and normalized to 500,000 cells. Supernatants were analyzed 48 h after the end of treatment. (**a**,**b**) Results are from three independent experiments (*n* = 3) and show the mean intensity value. (**c**,**d**) Results are from three independent experiments (*n* = 3); n.d., not detectable; ns, not significant; one-way ANOVA; mean ± s.d.; an asterisk (*) indicates a significant difference from the respective controls with *p* < 0.05.

**Figure 6 cells-11-01514-f006:**
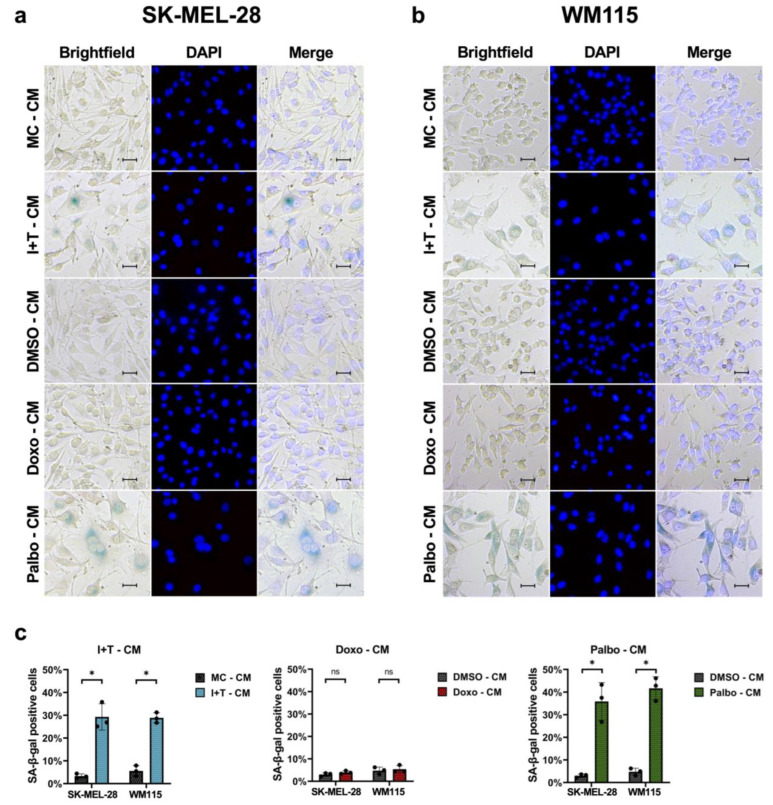
Induction of SA-β-gal activity after treatment with conditioned media (CM) derived from cytokine- and palbociclib-treated SK-MEL-28 and WM115 melanoma cells. (**a**) Representative images of SA-β-gal activity and DAPI staining in SK-MEL-28 (**a**) and WM115 (**b**) cells following treatment with CM for 96 h. Bars = 50 μm. (**c**) Quantification of SA-β-gal-positive cells. (**a–c**) Results are from three independent experiments (*n* = 3); ns, not significant; unpaired Student’s *t*-test; mean ± s.d.; an asterisk (*) indicates a significant difference from the respective controls with *p* < 0.05.

**Table 1 cells-11-01514-t001:** Cell numbers (×10^6^) that were counted and calculated in the growth arrest assay (Figure 2a,b) for each condition. The mean value (± s.d.) is indicated.

Cell Line	Passage	MC	I+T	DMSO	Doxo	Palbo
SK-MEL-28	−1	0.75 (± 0)	0.75 (± 0)	0.75 (± 0)	0.75 (± 0)	0.75 (± 0)
	0	6.90 (± 1.07)	0.73 (± 0.19)	6.05 (± 2.18)	2.78 (± 0.18)	1.05 (± 0.14)
	1	30.53 (± 4.51)	0.81 (± 0.13)	30.28 (± 5.49)	2.99 (± 1.82)	1.37 (± 0.27)
	2	260.41 (± 35.00)	1.12 (± 0.12)	244.28 (± 53.66)	3.66 (± 0.37)	1.84 (± 0.21)
WM115	−1	0.75 (± 0)	0.75 (± 0)	0.75 (± 0)	0.75 (± 0)	0.75 (± 0)
	0	8.07 (± 0.75)	1.46 (± 0.12)	7.39 (± 0.98)	1.41 (± 0.57)	1.04 (± 0.12)
	1	39.15 (± 1.89)	1.77 (± 0.15)	34.64 (± 2.26)	1.70 (± 0.77)	1.35 (± 0.37)
	2	232.01 (± 13.08)	2.07 (± 0.50)	237.77 (± 49.65)	1.94 (± 0.76)	1.64 (± 0.37)

## Data Availability

All data presented in this study are available in this article.

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
