# Peer review of "IFN-γ and TNF Induce Senescence and a Distinct Senescence-Associated Secretory Phenotype in Melanoma"

_cells, 2022, doi:10.3390/cells11091514_

Round 1

Reviewer 1 Report

Homann and colleagues proposed a research article aimed at investigating the role of different treatments in inducing a senescence-associated secretory phenotype in melanoma. For this purpose, the authors performed several in vitro experiments on two melanoma cell lines evaluating different molecular aspects. Overall, the manuscript is very interesting, however, I have some major concerns regarding the rationale of the study and the link with immunotherapy. Please address the minor/major revisions reported below:
1) In the title, please use the full form of SASP;
2) Why was the MTT assay not used for the evaluation of cell growth? Please clarify;
3) Throughout the manuscript there are some minor grammar errors. Please check carefully the entire text;
4) Both Introduction and Discussion sections should be shortened;
5) In the Introduction section, the authors widely described the association existing between immunotherapy, immune checkpoints and melanoma cell senesce. However, the entire manuscript is based on experiments performed using chemotherapeutic agents or selective inhibitors without considering the use of ICIs in animal models or evaluating the effects of these drugs on the immune cells. In addition, doxorubicin is not currently widely used in clinics as better results have been obtained through targeted therapy and immunotherapy. On these bases, the authors have to better explain (and experimentally demonstrate) the senescence-immune cell-immune checkpoints axis or clearly declare the limits of the study concerning this aspect;
6) In the Discussion section, the authors no longer describe the role of immunotherapy and immune cells in cell senescence. In the Introduction or Discussion sections, the authors should describe how immunotherapy has revolutionized the treatment of cutaneous melanoma improving patients’ survival and limiting the use of very toxic agents like doxorubicin. For this purpose, please see:
- PMID: 32582963
- PMID: 29567702
- PMID: 32671117

Reviewer 2 Report

The authors to show the relationship between cytokine treatment and senescence induction in melanoma. Authors conclude that senescence induction via cytokines may lead to a self-sustaining senescence surveillance in melanoma.

The discussion is very detailed and elaborates on possible advantageous as well as disadvantageous effects.

Major Points:

  • When reading the title and the abstract it is expected that the authors show consequences of senescence induction in melanoma on tumor associated immune cells. Unfortunately, this is not the case. Neither NK cells nor macrophages or T-cells are used for e.g. co-culture experiments. Authors are adviced either to rewrite misleading messages in the title and abstract or to conduct the corresponding experiments so that message and text go along together.
  • Melanoma cells are known to frequently carry CDKN2A deletions. Making them unsuitable for p16 and p14 analysis. Interestingly TP53 is most often WT. Hence, authors should include TP53 Westernblots in their results. In addition, two melanoma cells represent the bare minimum, more lines would be better.
  • Line 267-269, 450: To interpret nuclear morphology DAPI or similar staining needs to be shown.
  • 285, Figure 2: Whole presentation of data could be better – why does it look like the cell number is zero for both treatment and control at passage -1 and 0?
  • 323, Figure 3: The gates for the between control and treatment are set differently – they should be the same to compare the results. It would be nice if percentages for the gates were directly in the plots. Also, the colors for the percentages in d) are different than in the plots (S is darker in the plots, the other way around in the figure underneath).
  • In 347-348 it is stated that senescent cells primarily arrest in G1, but in figure d) and e) the percentages of cells in G1 only get more for palbo treatment compared to the control. Doxo and interferon do not increase G1% because subG1 is increasing.

Minor Points:

  • 76: “primarly” -> primarily
  • 114: are the 0,12% DMSO used as a control equal to the amount of the DMSO that was put on the cells through the doxo and palbo treatments?
  • 137, 143-144: “for each replicate, three wells per condition” -> for each condition, three replicates
  • 190: “cytokines or drugs, or controls” -> cytokines, drugs or controls
  • 241: generally, throughout the paper you change between “IFN-γ and TNF, doxorubicin, and palbociclib” and “IFN-γ and TNF, doxorubicin, or palbociclib” -> always use the same
  • 244: generally for all figures, why is there no further categorization into ns,*,**,***,… but just ns or * ?
  • 244: the gal staining looks very weak
  • 264: “remained stable” -> why is there a much higher signal for palbo in SK-MEL-28?
  • 288: “contro (a,b)l” -> control (a,b)
  • 325: histograms -> aren´t they plots?
  • 368-408: why talk about both the results without normalization and with normalization to the cell number – the normalized results are enough
  • 385: “Fig. 5a-d” – you have not talked about c) and d) at this point
  • 407: “significantly increased” – compared to what? The control, the other treatments?
  • 409-412: why do you even show IL-12 then? Why is there no signal if the ELISA detects both the p35 and p40?
  • 446-449: why didn´t you try this?
  • 484: do you mean IL-8? You never talked about IL-18 before
  • 484: “was” -> were
  • 527: “profinflammatory” -> proinflammatory
  • 529: “senescenct” -> senescent

Reviewer 3 Report

I have only one major concern with this paper, which pertains to the description of the LDH “cytotoxicity” assay and the Y axis of Fig. 1d as well as the corresponding text. In 2013, it was highlighted that treatment-induced injury to cancer cell membrane (resulting in LDH release; large dye uptake, etc) can be rapidly mended (PMID 2355892). Since then it has become evident that cancer cells can recover from brink of death (downstream of membrane injury in the process of apoptosis) through anastasis (see, e.g., PMID 34359573). Therefore, to assume that the 96 well plate assay used by Homann et al. (and thousands of other authors) measures "cytotoxicity" can be misleading. The assay measures changes in optical density of the medium overlaying cells in treated versus control wells. Therefore, the Y axis of Fig. 1d needs to be changed to something like “Absorbance 490-680 (nm)” (as is used in Figures of the Thermo Fisher website) or “LDH release (% ???)” but not cytotoxicity. The text corresponding to LDH release will also need to be changed to reflect what is measured.

Round 2

Reviewer 1 Report

The authors well-addressed almost all of my previous comments. However, they failed in addressing my concerns related to the relationship existing between immunotherapy and cell senescence. I understand that it is difficult and time-consuming to perform additional experiments, however, the authors have to provide some data about these relationships, even performing computational analyes. 

Reviewer 2 Report

The authors answered all open questions.

The manuscript is now significantly improved.

Author Response

Thank you very much for your constructive suggestions.

This manuscript is a resubmission of an earlier submission. The following is a list of the peer review reports and author responses from that submission.

Round 1

Reviewer 1 Report

In the article titled “Cytokine-induced senescence in melanoma cells is associated with a high proinflammatory secretome” the authors describe the pro-inflammatory feature of the SASP generated by cytokine-induced senescent melanoma cells. The topic of the article is interesting as cellular senescence is cutting-edge in tumor biology and is attracting the attention of the scientific community. Nevertheless, some points should be clearly addressed before publication, as listed below. The main reviewer’s concern regards the biological relevance of the study as only in vitro experiments with cell lines have been performed.

Major points:

  • The percentage of beta-Gal positive melanoma cells after I+T treatment is not high (it does not reach 40%, Fig. 1) and the staining of positive cells is not striking. For this reason, it should be mandatory to evaluate whether the pro-inflammatory cytokines released by the I+T treated cells come from the SASP of senescent cells or also from non-senescent ones responding to cytokine stimulation. Analyses at much earlier (24-48h) time points could help to clarify this issue (at early times the cellular senescence program should not be fully achieved and thus the SASP contribution little relevant).
  • In the Reviewer’s opinion, the G1 to G2 shift discussed by the authors is not supported by data as no significant difference is shown between G2 96 h sample versus G2 144h sample (Fig. 4). Please verify and explain better.

Minor points:

  • The quality of the figures is poor, please provide high resolution figures with higher font size.
  • In the abstract and introduction the link between ICB and cellular senescence is over-estimated and not clear. Only few and mostly self-citing references are provided (number 6-10) about this topic. Different mechanisms of action have been proposed for ICB therapy (ranging from unleashing T cell activity to Treg targeting), thus the statement at line 11-12 of the abstract is misleading. The cytokine-induced senescence is barely described by literature in the field. Please, be aware of the consensus to define cellular senescence (Vassilis Gorgoulis et al., Cell. 2019 Oct 31;179(4):813-827. doi: 10.1016/j.cell.2019.10.005; Hui-Ling Ou et al., Mol Oncol. 2021 Oct;15(10):2634-2671. doi: 10.1002/1878-0261.12807).
  • The authors should specify that TNF refers to TNF-alpha, if so.
  • There is a minor mistake in the sub-paragraph 10. Statistical analysis. “Statistically different” at line 198 should be modified in “statistically significant”.
  • There is a minor mistake at line 2010 “Furhter” should be modified in “Further”.
  • The authors should make clear the reason for adopting the reported TNF and IFN-gamma concentrations and whether those levels might be, in somehow, relevant in vivo / physio-pathological conditions. Furthermore, the authors should verify and state that no cell-death is observed at the indicated concentrations (sub-G1 analysis is insufficient).
  • Fig. 1. The authors should specify the number of cells/fields used for the quantitative analysis of SA-β-Gal positive cells.
  • Lines 228-230. The authors report that the growth arrest is stable for all the investigated conditions, but the palbociclib retention into the lysosome compartments and the following release after drug withdrawal was not considered (please see Llanos S et al., Oncogene. 2019 May;38(20):3886-3902. doi: 10.1038/s41388-019-0695-8.). A not stable cell cycle arrest could account for the not observed increased expression of p21 in palbociclib-treated cells.
  • Fig. 3. Is there any significant difference? In the Y axis should be clear that values refer to Ratio p16(or p21 or p27) / beta-actin versus MC.
  • In the discussion more recent references about cellular senescence are welcome, especially those regarding the immunological implications (e.g. Prasanna PG et al., J Natl Cancer Inst. 2021 Oct 1;113(10):1285-1298. doi: 10.1093/jnci/djab064; Fitsiou E et al., Semin Cancer Biol. 2021 Mar 26:S1044-579X(21)00071-7. doi: 10.1016/j.semcancer.2021.03.021; Cuollo L et al., Biology (Basel). 2020 Dec 21;9(12):485. doi: 10.3390/biology9120485; Antonangeli et al, J Leukoc Biol. 2019 Jun;105(6):1275-1283. doi: 10.1002/JLB.MR0718-299R). This may increase the interest of the readers of different backgrounds and contextualize the study in the state of the art.

Reviewer 2 Report

In this study Homman et al. compare melanoma senescent cell induced by chemotherapeutics and cytokines. The results are interesting, showing differences in cell cycle arrest and SASP composition depending on the nature and time of the induction stimuli. The manuscript is well written and of biomedical interest but some issues listed below preclude its publication in its current form. Special attention should be paid to the number of biological replicates and statistical analysis of the results.

  1. For the SA-beta –gal assays please include more information on the components of the kit and indicate at what pH was beta-galactosidase activity determined using X-gal.
  2. If possible, increase font size in the text and numbers of all the graphs in figures 2, 3, 4d, 4e; in particular for the graphs in figure 2.
  3. In figure 2b there appears to be a lot of dispersion in the values presented in the graphs? Are the levels of p21 significantly higher in I+T or Doxo treated cells than control cells as implied in the text?
  4. The results in figure 4 were performed in only 2 biological replicates. Please include one more (n=3) and perform statistic analysis of the results in order to determine if the differences obtained between conditions are significant.
  5. Please indicate the number of biological replicates in experiments presented in figure 5A and B, and indicate if the results were normalized using a reference gene such as actin.
  6. Please indicate the number of biological replicates in experiments presented in figure 5C and D, and indicate the number of cells in each condition.
  7. In figure 5E and F and in Figure A2, please indicate if the differences between treatments (Doxo vs I+T, Doxo vs Palbo, I+T vs Palbo) are statistically significant, as implied in the text.
  8. Figures 5E and F should be substituted by the ones presented in figure A2, in which cytokine values were normalized by cell number. Since, as the authors mention in the text, the number of cells in each condition is different in senescent and non –senescent cells.
